# Fully Atomistic Molecular Dynamics Computation of Physico-Mechanical Properties of PB, PS, and SBS

**DOI:** 10.3390/nano9081088

**Published:** 2019-07-29

**Authors:** Yang Kang, Dunhong Zhou, Qiang Wu, Fuyan Duan, Rufang Yao, Kun Cai

**Affiliations:** 1College of Water Resources and Architectural Engineering, Northwest A&F University, Yangling 712100, China; 2School of Water Resources and Hydropower Engineering, Wuhan University, Wuhan 430072, China; 3Centre for Innovative Structures and Materials, School of Engineering, RMIT University, Melbourne 3001, Australia

**Keywords:** molecular dynamics, glass transition, uniaxial tensile, linearized Eyring-like model, Young’s modulus

## Abstract

The physical properties—including density, glass transition temperature (*T*_g_), and tensile properties—of polybutadiene (PB), polystyrene (PS) and poly (styrene-butadiene-styrene: SBS) block copolymer were predicted by using atomistic molecular dynamics (MD) simulation. At 100 K, for PB and SBS under uniaxial tension with strain rate $\dot{\varepsilon }$ = 10^10^ s^−1^ and 10^9^ s^−1^, their stress–strain curves had four features, i.e., elastic, yield, softening, and strain hardening. At 300 K, the tensile curves of the three polymers with strain rates between 10^8^ s^−1^ and 10^10^ s^−1^ exhibited strain hardening following elastic regime. The values of Young’s moduli of the copolymers were independent of strain rate. The plastic modulus of PS was independent of strain rate, but the Young’s moduli of PB and SBS depended on strain rate under the same conditions. After extrapolating the Young’s moduli of PB and SBS at strain rates of 0.01–1 s^−1^ by the linearized Eyring-like model, the predicted results by MD simulations were in accordance well with experimental results, which demonstrate that MD results are feasible for design of new materials.

## 1. Introduction

To a great extent, the applications of polymers are determined by their physico-mechanical properties which are closely connected with the mechanical states at ambient temperature. Furthermore, the mechanical states of a polymer mainly depends on the relationship between its glass transition temperature (*T*_g_) and room temperature (*T*_r_), e.g., it is elastic *T*_g_ > *T*_r_, or vice versa, it is glassy [1]. Poly(styrene-butadiene-styrene: SBS) tri-block copolymers are one of the main families of thermoplastic elastomers [2]. The S comprises a thermoplastic phase (i.e., polystyrene: PS) with contents in a range from 10% to 40% [2], and B belongs to an elastomeric phase (i.e., polybutadiene: PB). SBS tri-block copolymers own a lot of distinctive mechanical properties by virtue of their microphase separation [3,4], which lead to their versatility in commercial applications [5]. Herein, the two phases PB and PS of homopolymers retain most of the physico-mechanical properties. For example, SBS consists of two values of *T*_g_, which illustrate the corresponding characteristics of PS and PB [6], respectively. Therefore, the PS phase behaves strong and rigid, whilst, the PB phase is soft and elastomeric at room temperature. In other words, it holds the characteristics both of rubber and plastic. Briefly, the three polymers of SBS, PB, and PS are typical thermoplastic polymers, and are chosen as model polymers in this study.

Usually, properties of prepared polymers can be precisely measured by experiments. However, the experimental tests are expensive and time-consuming. Moreover, the properties of newly developed polymers which are not prepared yet are not possible to achieve by experiment due to difficulty in observing the micro-structure development of the polymer, which directly determines the properties. Currently, molecular dynamics (MD) simulation approach is a potential way to overcome these difficulties. In an MD simulation, the movement of atoms and/or molecule can be predicted based on such physical principles as Newton’s law of motion, thermodynamics, and so on. Using the method, the material behaviors at atomistic level can be estimated. Until now, MD simulation has been widely applied to study the physical and mechanical properties—e.g., density [7,8], *T*_g_ [9], Young’s modulus [10]—of different polymers. It is worth noting that in studying the tensile properties of PB, PS, and SBS by using non-equilibrium MD (NEMD) method, high strain rate (e.g., from 10^8^ s^−1^ to 10^10^ s^−1^ in the present study) has to be adopted successfully to achieve large deformation of copolymers within acceptable/affordable simulation time. The loading rates in simulation are much higher than those applied in quasi-static mechanical experiments according to standard test methods like ISO, ASTM etc. with deforming rate of ~0.01 s^−1^. Thus, a linearized Eyring-like model [11,12,13] was chosen to extrapolate the Young’s modulus at experimental loading rates.

In the present work, the physico-mechanical of PB, PS, and SBS was evaluated by using a fully atomistic model, which would be verified by comparing the predicted densities and experimental results. Meanwhile, the values of *T*_g_ were calculated according to the relationship between the specific volumes and temperatures. Moreover, the mechanical properties of the systems under uniaxial tension were simulated. Decomposition of potential energy was fulfilled for illustrating the mechanisms of their deformation at different stages. Subsequently, stress–strain behaviors of PB, PS, and SBS and their strain-rate dependence were investigated by linearized Eyring-like model [11,12,13], whose purpose is to build the relationship between the experimental and the simulation data for potential applications, i.e., it means that the linearized Eyring-like model [11,12,13] is proper to extrapolate/compute the Young’s modulus at experimental loading rates. Namely, it could be used to compute/predict the virtual materials, which are not real but conceived.

## 2. Details of the MD Simulation and Method

### 2.1. Force Fields

In MD simulation, condensed-phase optimized molecular potentials for atomistic simulation studies (COMPASS) [7,8] force field was adopted. The force field is one of the first ab initio force fields. It sources from the polymer consistent force field (PCFF) [14,15], and was optimized by experimental data. The force field, which can accurately evaluate the interaction among a broad range of molecules and polymers, was often adopted to optimize and predict the structural and thermo-physical properties [16,17]. Thus, the force fields of COMPASS (density and *T*_g_) and PCFF (tensile properties) were chosen in the present simulations, and the total potential energy ($\Pi_{total}$) reads
(1)$$\begin{array}{c} \Pi_{total}=\sum_{b}\left[k_{2}{\left(b-b_{o}\right)}^{2}+k_{3}{\left(b-b_{o}\right)}^{3}+k_{4}{\left(b-b_{o}\right)}^{4}\right]\\ +\sum_{\theta }\left[k_{2}{\left(\theta -\theta_{o}\right)}^{2}+k_{3}{\left(\theta -\theta_{o}\right)}^{3}+k_{4}{\left(\theta -\theta_{o}\right)}^{4}\right]\\ +\sum_{\phi }\left[k_{1}\left(1-\cos\phi \right)+k_{2}\left(1-\cos2\phi \right)+k_{3}\left(1-\cos3\phi \right)\right]\\ +\sum_{\chi }k_{2}\chi^{2}+\sum_{b,b^{\prime }}k\left(b-b_{o}\right)\left(b^{\prime }-b_{o}^{\prime }\right)\\ +\sum_{b,\theta }k\left(b-b_{o}\right)\left(\theta -\theta_{o}\right)+\sum_{b,\phi }\left(b-b_{o}\right)\left[k_{1}\cos\phi +k_{2}\cos2\phi +k_{3}\cos3\phi \right]\\ +\sum_{\theta ,\phi }\left(\theta -\theta_{o}\right)\left[k_{1}\cos\phi +k_{2}\cos2\phi +k_{3}\cos3\phi \right]\\ +\sum_{b,\theta }k\left(\theta^{\prime }-\theta_{o}^{\prime }\right)\left(\theta -\theta_{o}\right)+\sum_{\theta ,\theta ,\phi }k\left(\theta -\theta_{o}\right)\left(\theta^{\prime }-\theta_{o}^{\prime }\right)\cos\phi \\ +\sum_{i,j}\frac{q_{i}q_{j}}{r_{ij}}+\sum_{i,j}\varepsilon_{ij}\left[2{\left(\frac{r_{ij}^{o}}{r_{ij}}\right)}^{9}-3{\left(\frac{r_{ij}^{o}}{r_{ij}}\right)}^{6}\right] \end{array}$$

The energy terms in Equation (1) are classified into two categories. One is the group of bonding interactions in terms of bond length (*b*), bond angle (*θ*), dihedral angle (ϕ), improper term (χ), and their coupling. The other is the non-bond interaction, including the Coulombic electrostatic force and van der Waals (vdW) force. The subscripts *i* and *j* refer to the pair of atoms with distance of *r_ij_*. Moreover, the vdW interactions are estimated by the Lennard-Jones (LJ 9-6 model) potential function with parameters *_ij_* and $r_{ij}^{o}$.

### 2.2. Constructing the Polymeric Model

The molecular models in Figure 1 were built on the ‘Amorphous Cell’ package in Materials Studio (Accelrys Inc., San Diego, CA, USA) [18]. The monomer structures of PB, PS, and SBS were built on the ‘Build Module’, and the ‘Homopolymer Builder Module’ was used to create the homopolymer chains of PB and PS. The block copolymer of SBS was created on the ‘Block Copolymer Builder Module’. As shown in Figure 1, which display the bottom-up process from monomers, repeat units, the chains of polymer to the molecular model. For the block copolymer of SBS (Figure 1c), it contains 8 hard segments (PS) and 25 soft segments (PB), i.e., the ratio of m/n = 8/25, and the segments ratio of the weight percent is PS/PB = 31/69. To obtain the models of PB, PS, and SBS after building single polymer chains, a predetermined number of polymer chains (in Table 1) were randomly packed into a cell with a low density of 0.6 g/cm^3^ on the ‘Amorphous Cell Module’ package. The cell has three-dimensional periodic boundary conditions. Finally, the initial configurations of PB, PS, and SBS models (Figure 1j–l) were obtained. Details of the three cells of MD simulation system were listed in Table 1.

### 2.3. Methodology of MD Simulation

After obtaining the initial configurations of PB, PS, and SBS, their geometry was optimized by using the ‘Forcite Module’ to eliminate the unreasonable contacts. Furthermore, obtaining stable configurations—i.e., with the minimal potential energy—simulated annealing processes for them were performed by growing temperature first from 300 K to 600 K and then dropped back to 300 K. After simulated annealing, each cell was simulated in the NPT (constant number (N) of particles, constant pressure (P), and constant temperature (T)) ensemble at 298.15 K and 1 atm pressure until the system was fully relaxed and stable. Subsequently, a series of NPT MD simulations were fulfilled to obtain the desired properties by using the temperature cycle protocol in an increment of 25 K from 600 K to 25 K (i.e., the cooling down of the systems is 25 K/1000 ps). Note that the configuration used for each subsequent temperature is a balanced conformation at the previous temperature. Therein, the Nosé–Hoover thermostat and Berendsen barostat algorithms were adopted to control the temperature and pressure, respectively. The non-bond interactions—i.e., vdW and electrostatic interactions—were evaluated the group-based summation and Ewald summation methods, respectively. Periodic boundary conditions on a simulation box and a cutoff radius of 1.55 nm were applied. In each simulation, timestep for integration was set to 1 × 10^−6^ ns.

To investigate the mechanical responses of the stable structures, NEMD simulation was carried out to show the deformation under uniaxial tension by using the Large Scale Atomic/Molecular Massively Parallel Simulator (LAMMPS, Sandia National Laboratory, USA) [19]. The equilibration configurations were first quadrupled and then relaxed adequately. For example, the system was first relaxed at NPT ensemble at 600 K for 1 ns, then cooled to the desired temperature (e.g., *T* = 300 K) within 1 ns, and finally relaxed for another 1 ns at the desired temperature. In relaxation, the stress–strain was calculated [20]. A constant extension strain rate was applied along a dimension of the cell at a constant temperature. The strain rates, $\dot{\varepsilon }$, can be 10^10^, 10^9^, 10^8^, 10^7^ per second and so on, which have already been used in previous studies [20,21].

## 3. Result and Discussions

### 3.1. Model Verification and Density Prediction

As shown in Figure 2a,b, both the total potential energy and non-bonded energy fluctuate slightly after 0.3 ns of relaxation. Hence, after 0.5 ns, all of the cells have been in the thermodynamic equilibrium state. Meanwhile, in Figure 2c the temperature fluctuates with amplitude no more than 10 K (about ±3.35% of relative error), which means the temperature of system was controlled very well. Hence, it is concluded that the models were valid. As presented in Figure 2d, the predicted densities changed drastically in the start of simulation, and then varied slightly after about 0.5 ns of relaxation. Accordingly, the stable values of the predicted densities of PB, PS, and SBS are listed in Table 2 at at *T* = 298.15 K and *P* = 1 atm. Compared to the experimental data, the relative errors are of 0.33%, 2.29%, and 0.75% for PB, PS, and SBS, respectively. They are within acceptable error ranges [22]. This high agreement demonstrates that the validity of the present configurations, force field, and the simulation procedure.

### 3.2. Glass Transition Temperature (T_g_)

Figure 3 shows the relationships between the specific volumes and temperatures for PB, PS and SBS systems. These data were obtained from several results of MD replicas. Using piecewise fitting, the values of *T*_g_ in different temperature region were estimated. For example, *T*_g_ = 179 K for pure PB (Figure 3a). Fortunately, the value of *T*_g_ is slightly different from those obtained by other researchers (185 K) [27,28], and is only 9 K higher than the experimental value (170 K) [29,30]. Similarly, for the PS system (Figure 3b), the value of *T*_g_ was 380 K, given the vastly different employed cooling rates, which is compatible with the temperatures obtained by experiments (353–383 K) [31,32,33,34]. These slight differences between simulations and experiments are mainly originated from the differences between simulations and experiments at cooling rates [35] and molecular weights [34]. It demonstrated that on the nanoscale, the values of *T*_g_ predicted by MD simulations match well with those by experiments. This is another evidence to indicate that the present model and techniques in MD simulations are available for predicting the real copolymer system.

It is well known that SBS exhibits two glass transitions with two different transition temperatures. One is determined by the PS block of SBS, i.e., a thermoplastic block, and is labeled as *T*_g2_, which is higher than room temperature. The other depends on the PB block of SBS—i.e., a rubber block—and is labeled as *T*_g1_, which is below room temperature [6]. As shown in Figure 3c, in the butadiene domain *T*_g1_ = 189 K was higher than *T*_g_ = 179 K for a pure PB. This conclusion has been verified by experiments [36]. Meanwhile, *T*_g2_ = 346 K for the PS domain in SBS copolymer was above room temperature. It has already been underpinned by experimental findings [36,37]. Note that the value of *T*_g2_ was significantly lower than *T*_g_ = 380 K due to two reasons. One is that the degree of polymerization of PS in SBS is lower, and the other is that the soft segment PB is connected to the hard segment PS, which increases the movement ability of hard segment PS segment in SBS [31,37]. Comparison between the simulation and experimental results on glass transition temperatures of PB, PS, and SBS were listed in Table 3.

### 3.3. MD Computation of the Mechanical Properties

#### 3.3.1. Internal Energy Evolution

Figure 4 shows the normalized total potential energy and the normalized individual components of the total potential energy of PB, PS, and SBS systems deformed at 100 K for strain rates of 10^10^ s^−1^ and 10^9^ s^−1^ during the tensile processes. It can be seen that at different strain rates, all the normalized energies evolve with the similar tendencies, respectively, while, when the strain rate is lower, the normalized total energy and the normalized individual components of the energy become lower. The reason is that the lower strain rate will offer more time for polymer chains to disentangle as a way to accommodate deformation. For glassy PB, PS, and SBS, initially, the normalized total potential energy increased sharply. One can also find that the slope of normalized non-bond energy—i.e., vdW interaction between polymer chains—almost increase synchronously with the normalized total potential energy, while slopes of other normalized energies keep almost constant. The normalized non-bonding energy was generated during unwrapping of the polymer entangled strands, which had initially coil-like configurations, in the elastic regime (ca. ε < 0.1). It reveals the mechanism of chain slippage during elongation. After the elastic regime, for glassy PB, PS, and SBS, the slopes of normalized non-bond energy, bonding energy, and dihedral energy are gradually decreasing to about zero. It is worthy of noticing that the slopes’ absolute values of normalized angle energies increase dramatically for different polymers—i.e., for glassy PB and SBS—they keep fairly non-zero constant, while, for glassy PS, it stays zero (ca. ε > 0.3). Thus, we suspect that they may be in different physical mechanics states.

#### 3.3.2. Stress–Strain Curves

As plotted in Figure 5a or Figure 5c, the tensile curves at 100 K are divided into several regimes. In elastic regime, i.e., regime I, the stress increased approximately linearly with strain. After the yield point, the stress decreased continuously with strain (regime II), indicating PB and SBS are in the state of strain-softening, which is attributed to the drops of the intermolecular interactions (i.e., non-bonging energy) contribution to stress (Figure 4a,b,e,f, ε < 0.1). However, after the stress approaching the local minimum, further deformations of PB and SBS demonstrate the two polymers are in the state of strain-hardening (regime III), which is attributed that the intramolecular interactions, especially the angle energy contribution to stress is dominated (Figure 4a,b,e,f, ε > 0.3). These phenomena have been observed in real experiments [40,41], MD tension simulation [42], and Monte Carlo simulations [43]. However, compared with PB (Figure 5a) and SBS (Figure 5c) systems, there is no stress-hardening regime in the constitutive curves with respect to PS (Figure 5b), and the possible reason is that partial fracture occurred in the PS system because the slopes of normalized angle, bond, and dihedral energies keep almost zero while the slopes of non-bond energies increase slightly, i.e., non-bond energy increase while other energies keep constant (Figure 4c,d), ε > 0.3)). To verify the partial fracture occurred in the PS system, the snapshots of PS system under different strains (Figure 5d) are abstracted. That is, the PS had no defect at ε ≤ 0.2, which implies that the PS system was in an elastic state. However, at ε ≈ 0.41, the PS chains were partly separated under stretching, and the void in the simulation box occurred on both sides of the PS model due to slippage of the PS chains. With the strain increasing to some extent, e.g., ε = 0.92, most of the PS chains were tightened to be almost straight, which indicate that the entangled PS strands were separated, which implies part of the non-bonded interaction disappeared, i.e., the plastic deformation appeared at some point, or fracture in another word. Notably, for PB, PS, and SBS at 100 K, when considering the slope of a curve as the Young’s modulus under the corresponding loading rate, showing no obvious strain rate effects of Young’s modulus (i.e., the elastic regime (I, ε < 0.1)). Similar results have been reported [44,45,46].

#### 3.3.3. Young’s Modulus

In this work, elastic moduli of the copolymers are the essential of the mechanical properties, and were discussed below.

As indicated in Figure 6, at ambient temperature (i.e., *T* = 300 K), in regime I, the stress–strain curves were almost linear. Its gradient can be considered as the elastic modulus, i.e., Young’s modulus *E*, which is defined as
(2)$$E=\underset{\lambda \to 1}{\lim}\;\left(\frac{\sigma_{e}}{\lambda -\lambda^{-2}}\right)$$
where *λ* = 1 + ε_e_ is the tensile stretch ratio.

For PB (Figure 6a) and SBS (Figure 6c), the values of Young’s moduli at 300 K are dependent on strain rates seriously. However, for PS (Figure 6b), below strain value of ca. 0.5%, the stress–strain curves are nearly completely overlapped for all of strain rates applied, showing that in the linear elastic region no plastic deformation occurs and the elastic property of the PS is insensitive to strain rate, i.e., the initial elastic regime is independent of strain rates. The reason is that PS (*T*_g_ = 380 K) behaves thermoplastic, while PB (*T*_g_ = 179 K) and SBS (*T*_g1_ = 189 K, *T*_g2_ = 346 K) are elastomeric at 300 K. In other words, the difference of mechanical state among different polymers leads to their different mechanical responses.

Although the simulated systems in the present study were large enough from molecular dynamics viewpoint, the stress–strain responses shown in Figure 6 cannot be compared directly with the mechanical responses of real bulk elastomers (For PS, the initial elastic regime is independent of strain rates. And the calculated value of the PS’s Young’s modulus is ca. 3.2 ± 0.4 GPa, which is in good accordance with the experimental results (3.0 GPa [47], 3.4 GPa [48], 3.2–3.4 GPa [49])). The reason is that the strain rates in MD simulations are at least five orders of magnitude higher than those in experiments (0.01–1 s^−1^) (ISO 37, ISO 527, ASTM D412, ASTM D882, ASTM D638, GB/T 528, GB/T 1040, etc.). Therefore, a linearized Eyring-like model [11,12,13] was selected to extrapolate the Young’s Modulus at experimental loading rates. The model [11,12,13] is given as
(3)$$\dot{\varepsilon }={\dot{\varepsilon }}^{*}\exp\left(\frac{\sigma \cdot \upsilon^{*}}{2k_{B}\cdot T}\right)$$
where,
$\dot{\varepsilon }$
$\dot{\varepsilon }$^*^, *σ*, *υ*^*^, and *k*_B_ are the strain rate, the characteristic strain rate, applied stress above *T*_g_, an arbitrary activation volume, and Boltzmann constant, respectively. In general, the model is available for treating amorphous polymers under uniaxial compressive loading with yield stress, it is also available when considering elastic deformation under uniaxial tension as the initial modulus of elastomers [13]. The extrapolated results were presented in Figure 7.

As shown in Figure 7, for PB, the values of Young’s moduli predicted by MD simulation are consistent with those given by experiments (0.4 MPa [50], 0.7 ± 0.1 MPa [51], 1.13 ± 0.11 MPa [52], and 1.38 MPa [53]). Meanwhile, the extrapolated Young’s modulus of SBS system is also in good accordance with the experimental results (2.5 ± 0.1 MPa [54], 4.14 ± 1.61 MPa [39], and 4.1 MPa [55]). Hence, we can state that the MD simulation results are reliable for predicting the mechanical properties of copolymers according to the coincidence of the results of simulations and experiments.

## 4. Conclusions

In summary, we studied the physico-mechanical properties of amorphous polymers PB, PS, and SBS using a fully atomistic molecular dynamics simulation approach. The computational results were compared with those given by the experiments. Accordingly, major conclusions are summarized as follows:(1)The predicted densities of the PB, PS and SBS systems at *T* = 298.15 K and *P* = 1 atm match well with those obtained by experiments, and their densities are ca. 0.897 ± 0.0033 g/cm^3^, 1.021 ± 0.0037 g/cm^3^ and 0.933 ± 0.0034 g/cm^3^, respectively.(2)The glass transition temperatures of PB, PS and SBS were predicted based on the relationship between the specific volumes and temperatures. The predicted values of *T*_g_ of PB, PS and SBS are ca. 179 K, 380 K, 189 K and 346 K, respectively.(3)At 100 K, the tensile curves for PB and SBS systems at the strain rate of 10^10^ s^−1^, 10^9^ s^−1^ show four features, which have been observed in experiments, i.e., elastic, yield, softening and hardening.(4)At 300 K, for PB, PS and SBS under uniaxial loading conditions with the stress–strain responses being calculated in a range of strain rate from 10^8^ to 10^10^ s^−1^, their Young’s moduli were in good agreements with those given by experiments via extrapolation based on the linearized Eyring-like model.

## Figures and Tables

**Figure 1 nanomaterials-09-01088-f001:**
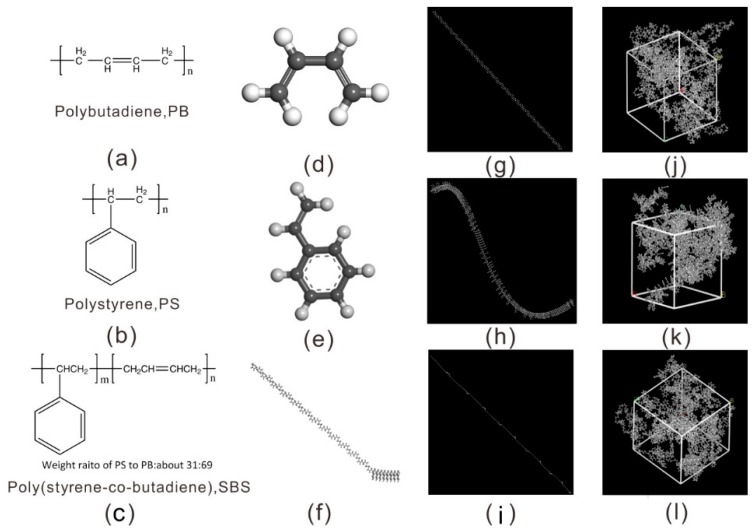
Detailed molecular structures and models of the three polymers. The schematic chemical structure for: (**a**) *cis*-1, 3-polybutadiene, PB; (**b**) polystyrene, PS; (**c**) styrene-butadiene-styrene block copolymer, SBS. The repeat units of molecular chain structures: (**d**) 1,3-butadiene, (**e**) styrene, (**f**) styrene-butadiene. Molecular chain configurations: (**g**) polybutadiene, (**h**) polystyrene, (**i**) styrene–butadiene–styrene block copolymer. The molecular configurations of the three different packing models: (**j**) PB, (**k**) PS, (**l**) SBS, respectively.

**Figure 2 nanomaterials-09-01088-f002:**
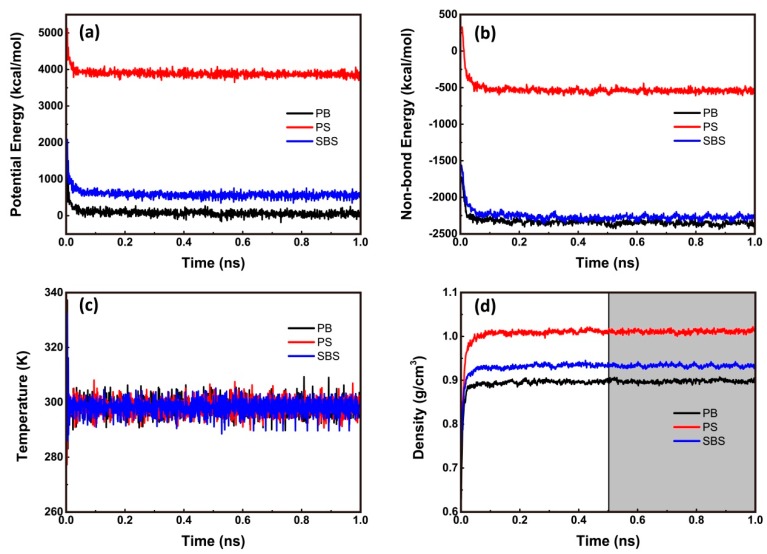
Historical curves of physical properties of systems. (**a**) Total potential energy (i.e., Equation (1)) (kcal/mol), (**b**) non-bonded energy (kcal/mol), (**c**) temperature (K), and (**d**) density (g/cm^3^). The mean values of density of PB/PS/SBS were obtained between 0.5 ns and 1.0 ns (the grey region)) at T = 298.15 K and P = 1 atm.

**Figure 3 nanomaterials-09-01088-f003:**
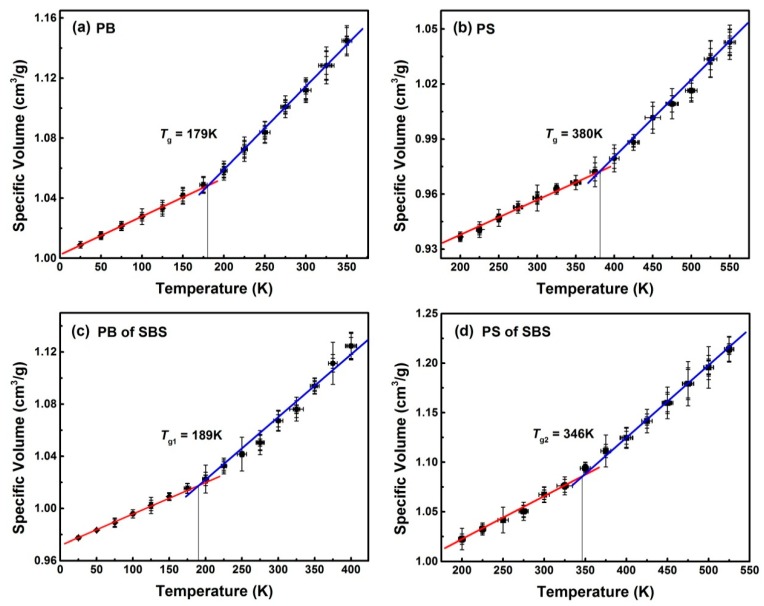
Temperature dependence of the specific volume (the reciprocal of density) for (**a**) PB system, (**b**) PS system, (**c**) PB of SBS system, and (**d**) PS of SBS system. Solid lines are the linear fits of the simulation data and error bars are the related standard deviations.

**Figure 4 nanomaterials-09-01088-f004:**
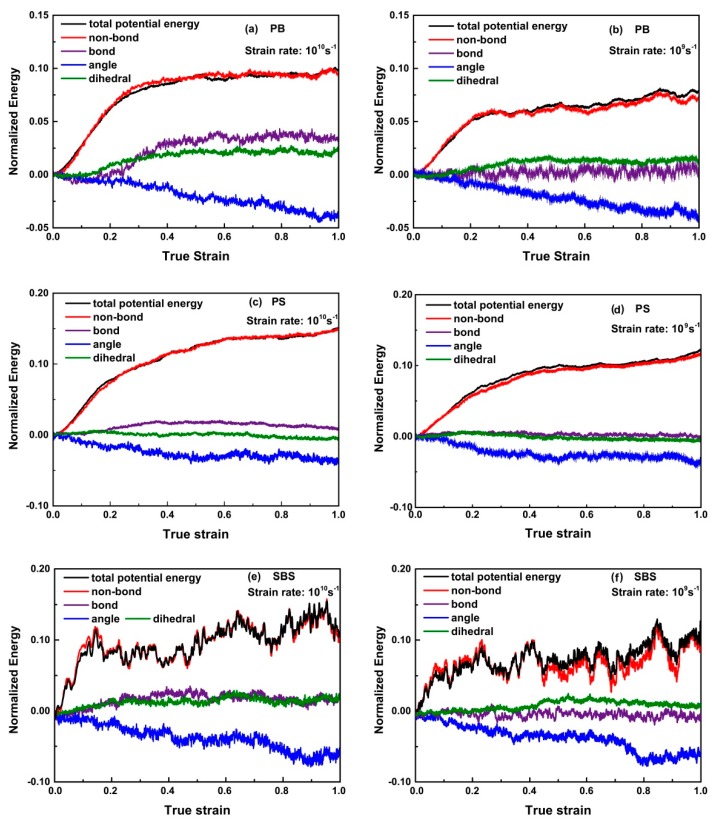
Evolution of normalized energy for: (**a**,**b**) PB; (**c**,**d**) PS; and (**e**,**f**) SBS at 100 K with strain rates of $\dot{\varepsilon }$ = 10^10^ s^−1^ and $\dot{\varepsilon }$ = 10^9^ s^−1^. Herein, the total potential energy contains for items, i.e., the non-bond energy, bond length energy, bond angle energy, and dihedral energy.

**Figure 5 nanomaterials-09-01088-f005:**
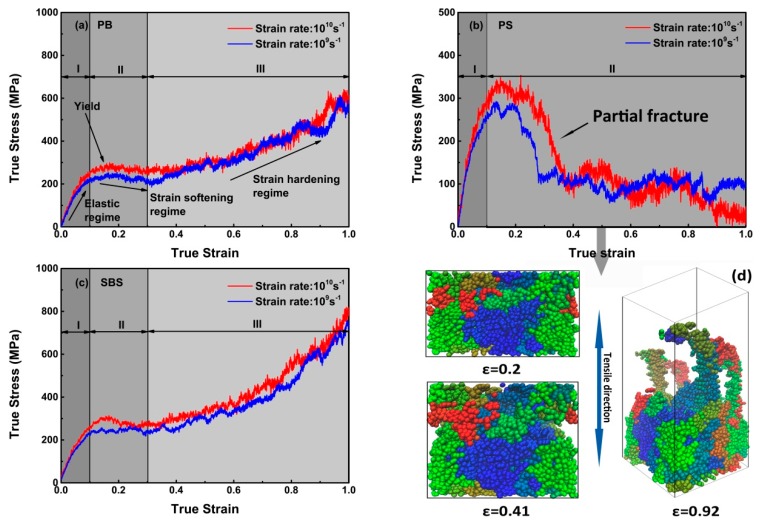
Stress–strain response of the polymer cell under uniaxial tension with strain rates of $\dot{\varepsilon }$ = 10^10^ s^−1^ and 10^9^ s^−1^ at 100 K. (**a**) PB; (**b**) PS; (**c**) SBS, respectively. The snapshots in (**d**) are obtained from PS system with different strains (ε). The PS chains are in different colors.

**Figure 6 nanomaterials-09-01088-f006:**
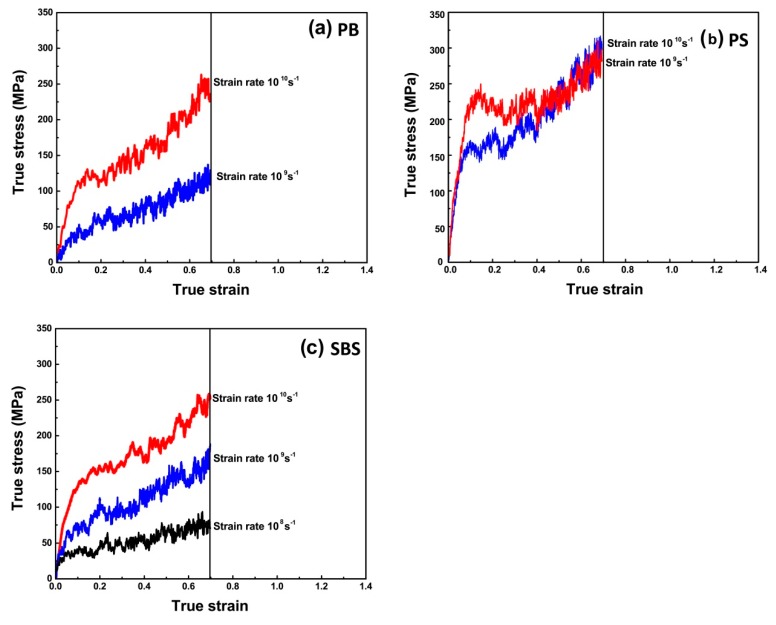
True stress–strain responses of (**a**) PB, (**b**) PS and (**c**) SBS at different strain rates $\dot{\varepsilon }$ and *T* = 300 K. Therein, the true stain and stress are defined as ε_t_ = *L_n_* (1 + ε_e_) and *σ*_t_ = *σ*_e_ (1 + ε_e_), respectively. *σ*_e_ is the engineering stress, ε_e_ is the engineering strain.

**Figure 7 nanomaterials-09-01088-f007:**
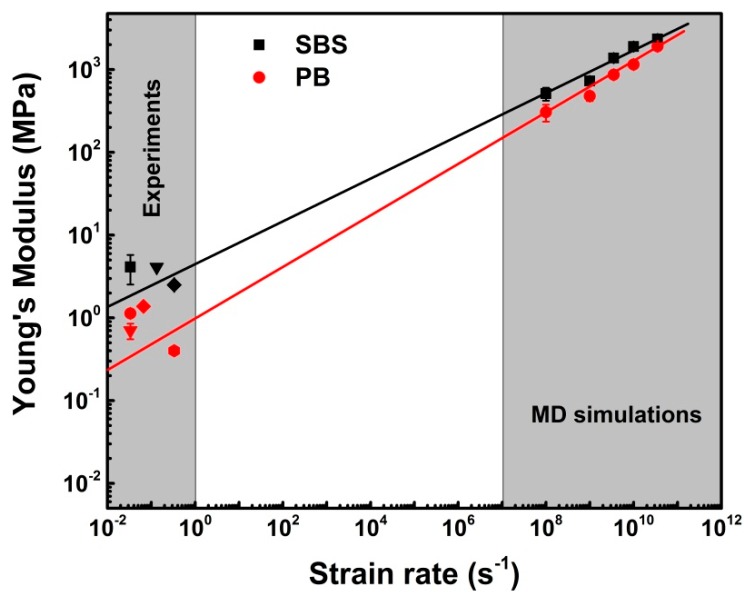
Young’s modulus (experiments and MD simulations) verse strain rate for PB and SBS systems at 300 K and 1 atm. Therein, the solid lines are obtained by fitting the data using Equation (3), and the error bars are associated for different loading conditions to ensure statistical sampling.

**Table 1 nanomaterials-09-01088-t001:** Details of the constructed initial models.

Polymers (-)	Chain No. (-)	Repeat Unit (-)	Molecular Formula (-)	Atom No. (-)	Cubic Cell Length (Å)
PB	6	120	2880C + 4332H	7212	47.596
PS	4	120	3840C + 3848H	7688	51.724
SBS	2	10	3280C + 4284H	7564	49.458

**Table 2 nanomaterials-09-01088-t002:** Densities of PB, PS, and SBS system from simulations and experiments at *T* = 298.15 K, *P* = 1 atm.

Polymer	Density ρ (g/cm^3^)		Error (%)
	Simulation	Experiment	
PB	0.897 ± 0.0033	0.90 [23]	0.33
PS	1.021 ± 0.0037	1.03–1.07 [24,25]	2.29
SBS	0.933 ± 0.0034	0.94 [26]	0.75

**Table 3 nanomaterials-09-01088-t003:** Glass transition temperatures (*T*_g_) by MD simulations and experiments.

Polymer	Glassy Transition Temperature *T*_g_ (K)	
	Simulation	Experiment
PB	179	182–210 [29,30]
PS	380	353–383 [31,32,33,34]
SBS	189 ^a^/346 ^b^	181 ^a^/340 ^b^ [38,39]

Note: ^a^ the soft/PB block of SBS, ^b^ the hard/PS block of SBS.

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
