# Peer review of "Fully Atomistic Molecular Dynamics Computation of Physico-Mechanical Properties of PB, PS, and SBS"

_nanomaterials, 2019, doi:10.3390/nano9081088_

Round 1
Reviewer 1 Report
The manuscript reports MD simulations of PB, PS and SBS with the aim of characterizing mechanical properties of these polymers. I have two serious concerns which may prevent publication of the manuscript, one technical and the other on the choice of the journal.
The former regards the mechanical behavior of two polymers that at 300 K are in completely different states, that is, PS is in the glassy state, while PB is the rubbery state. In turn, completely different mechanical properties are expected. Instead, qualitative comparison between the two polymers in Figure 6 indicates that the mechanical response of these two polymers is very similar. Also the case of the Young’s modulus (Figure 7) is suspicious. Glassy polymers exhibit values several orders of magnitudes larger than those of rubbery polymers. However in the present study SBS, which presents significant amount of rigid PS, exhibits similar Young’s modulus as that of PB. How can this outcome be explained within any physically sound framework?
Regarding the choice of the journal, I see no evident reason why the simulated properties of bulk polymers (periodic boundary conditions are chosen) would fit the scope of a journal such as Nanomaterials. If the size of segregated domain in SBS were in the nanometer scale, this could be a justification. However, the authors should provide convincing arguments that such alleged nanometers scale has significant impact on properties.
Apart from these I have the following minor comments:
1. The comparison between Tg(s) of the present study and experimental ones must account for the molecular weight and, overall, the vastly different cooling rates. For instance, in the case of PS of the present study with Mw = 12000 g/mol, Flory, Journal of Applied Physics 21, 581 (1950), reported an experimental Tg of 353 K. This is compatible with the 380 K Tg of the present study, given the vastly different employed cooling rates. However, the discussion in lines 170-180 is misleading and needs to be completely rewritten in line with my comment.
2. Figures 4 and 5 show data at 100 K, while Figures 6 and 7 do at 300 K. Why choosing two such different temperatures?
3. Quiescent properties at room temperature, after a short transient, are shown to be time invariant. However, once again, PS is glassy and therefore may undergo structural relaxation that results in time dependent properties. Have the authors accounted for this?
Reviewer 2 Report
This a decently written report of the application of standard MD atomistic simulations to characterize properties of polymeric aggregates. I have no particular objection to the content of the paper, the methods described and the results obtained, save that they are hardly original and that there are plenty of analogous works in the literature.(a search on web of science with keywords "molecular dynamics" and "polymers" gives 34271 results; of these, 620 refer to polybutadiene specifically).
The introduction contains an initial discussion about MD methods etc. which is essentially useless (eq. 1), being the typical subject of any worthy basic course in computational chemistry.
All together, the paper can be of interest for the readers of Nanomaterials but it hardly adds anything to the already vast literature in the field.
Author Response
Response to Reviewer 2 Comments
Point 1: This a decently written report of the application of standard MD atomistic simulations to characterize properties of polymeric aggregates. I have no particular objection to the content of the paper, the methods described and the results obtained, save that they are hardly original and that there are plenty of analogous works in the literature.(a search on web of science with keywords "molecular dynamics" and "polymers" gives 34271 results; of these, 620 refer to polybutadiene specifically).
The introduction contains an initial discussion about MD methods etc. which is essentially useless (eq. 1), being the typical subject of any worthy basic course in computational chemistry.
All together, the paper can be of interest for the readers of Nanomaterials but it hardly adds anything to the already vast literature in the field.
Response 1: Thanks to the reviewer for his/her time.
Yes, there has been a great deal of research on the properties of polymers by using molecular dynamics. However, this paper further demonstrates that simulations can predict data consistent with experimental data by the linearized Eyring-like relationship, which means that the linearized Eyring-like model is proper to extrapolate/compute the Young’s modulus at experimental loading rates. Therefore, it could be used to compute/predict the virtual/ conceived materials, which are not real prepared yet.

Round 2
Reviewer 1 Report
Altogether, the revised version of the manuscript and the response to the concerns raised in my previous report can be considered satisfactory.
From my viewpoint the problem of the suitability of the present manuscript for a journal like Nanomaterials still persists. The reason is that anyway the authors use their MD simulations to determine bulk properties. In other words, the modification of properties at the nanoscale is irrelevant for the present work. This concern, however, depends on the editorial policy and, thereby, goes beyond the scientific assessment of the present work.
Author Response
Thank the referees for their expert and responsible reviews. Thanks to the editor and assistant editor for his/her time.
Reviewer 2 Report
No comment
Author Response

(The authors gave the same response as above.)
